# Genome-Wide Association Study of Beta-Blocker Survival Benefit in Black and White Patients with Heart Failure with Reduced Ejection Fraction

**DOI:** 10.3390/genes14112019

**Published:** 2023-10-28

**Authors:** Jasmine A. Luzum, Alessandra M. Campos-Staffico, Jia Li, Ruicong She, Hongsheng Gui, Edward L. Peterson, Bin Liu, Hani N. Sabbah, Mark P. Donahue, William E. Kraus, L. Keoki Williams, David E. Lanfear

**Affiliations:** 1Department of Clinical Pharmacy, University of Michigan College of Pharmacy, Ann Arbor, MI 48109, USA; camposa@umich.edu; 2Center for Individualized and Genomic Medicine Research (CIGMA), Henry Ford Health System, Detroit, MI 48202, USAdlanfea1@hfhs.org (D.E.L.); 3Department of Public Health Sciences, Henry Ford Health System, Detroit, MI 48202, USA; jia.li.statistics@gmail.com (J.L.);; 4Heart and Vascular Institute, Henry Ford Health System, Detroit, MI 48202, USA; hsabbah1@hfhs.org; 5School of Medicine, Duke University, Durham, NC 27710, USAwilliam.kraus@duke.edu (W.E.K.)

**Keywords:** beta-blocker, heart failure with reduced ejection fraction, survival, genome-wide association study

## Abstract

In patients with heart failure with reduced ejection fraction (HFrEF), individual responses to beta-blockers vary. Candidate gene pharmacogenetic studies yielded significant but inconsistent results, and they may have missed important associations. Our objective was to use an unbiased genome-wide association study (GWAS) to identify loci influencing beta-blocker survival benefit in HFrEF patients. Genetic variant × beta-blocker exposure interactions were tested in Cox proportional hazards models for all-cause mortality stratified by self-identified race. The models were adjusted for clinical risk factors and propensity scores. A prospective HFrEF registry (469 black and 459 white patients) was used for discovery, and linkage disequilibrium (LD) clumped variants with a beta-blocker interaction of *p* < 5 × 10^−5^, were tested for Bonferroni-corrected validation in a multicenter HFrEF clinical trial (288 black and 579 white patients). A total of 229 and 18 variants in black and white HFrEF patients, respectively, had interactions with beta-blocker exposure at *p* < 5 × 10^−5^ upon discovery. After LD-clumping, 100 variants and 4 variants in the black and white patients, respectively, remained for validation but none reached statistical significance. In conclusion, genetic variants of potential interest were identified in a discovery-based GWAS of beta-blocker survival benefit in HFrEF patients, but none were validated in an independent dataset. Larger cohorts or alternative approaches, such as polygenic scores, are needed.

## 1. Introduction

Heart failure (HF) is a global pandemic that affects at least 26 million individuals [1]. Despite improvements in HF therapies over the past few decades, the one-year mortality rate is still approximately 30% globally [1]. Beta-blockers are one of the few pharmacologic therapies that can significantly reduce morbidity and mortality in patients with heart failure with reduced ejection fraction (HFrEF) [2]. Thus, beta-blockers are a cornerstone for guideline-recommended HFrEF therapy [2]. The landmark beta-blocker clinical trials for HFrEF demonstrated significantly improved outcomes, on average, in a large patient sample [3,4,5]; however, a substantial portion of HFrEF patients did not receive any benefit. Only ~22% of HFrEF patients demonstrated a marked and sustained improvement in their left ventricular ejection fraction (LVEF) with beta-blocker therapy [6,7]. In a randomized, double-blind clinical trial of HFrEF patients, the mean ± sd of the change in LVEF, after approximately 12 months of treatment, was +10.9% ± 11.0% and +7.2% ± 7.7% for carvedilol and metoprolol, respectively [8]. With the global prevalence of HF continuing to rise, the need to identify HFrEF patients that are most likely to benefit from this potentially life-saving therapy is of paramount importance.

Unfortunately, clinical characteristics do not predict which HFrEF patients will benefit from beta-blocker therapy. The clinical subgroup analyses of beta-blocker clinical trials showed similar benefit regardless of a patient’s age, sex, comorbidities, or HFrEF etiology [3,4,5]. More recent studies have shown that the benefits of beta-blockers are similar, regardless of a patient’s race [9]. Therefore, research efforts have focused on identifying novel factors that may predict beta-blocker benefits in HFrEF patients, such as genetics [10,11]. Pharmacogenetic studies of beta-blockers in HFrEF have primarily used the candidate gene approach (i.e., only a few genes are selected for analysis in the genome, based on a priori hypotheses that genes will play a role). Beta-blockers inhibit the neurohormonal drivers of HF pathophysiology via the sympathetic adrenergic and renin–angiotensin–aldosterone systems [12,13,14,15]. Accordingly, pharmacogenetic studies of beta-blockers in HFrEF patients have primarily focused on candidate genetic variants from these systems [10,11]. However, the results from candidate gene association studies have been inconsistent [10,11], which negates clinical utility, and suggests that additional and/or alternative genetic variants may be involved. The major limitation of candidate gene studies is that they can miss thousands of other genes in the genome, which may play a role. However, a genome-wide association study (GWAS) of beta-blocker survival benefit in patients with HFrEF has not previously been conducted. Therefore, the objective of this study was to use a GWAS to discover and validate novel genetic variant(s) that significantly interact with beta-blocker survival benefit in patients with HFrEF.

## 2. Materials and Methods

### 2.1. Discovery Dataset

The discovery dataset came from a prospective HF pharmacogenomic registry (HFGR) designed to discover novel ways to predict HF prognoses and responses to HF therapies [9]. The registry began enrolling patients in October 2007, within the Henry Ford Health System in Detroit, MI, USA, and enrolment was completed in March 2015. The inclusion criteria were as follows: age ≥ 18 years; a diagnosis of HF using the Framingham Heart Study criteria [16]; and at least one documented LVEF < 50% measured via echocardiography, nuclear stress tests, or radionuclide blood pool imaging. The LVEF cutoff was chosen to reflect patients with systolic HF because the study was designed and started prior to more recent reclassifications, which suggest that HFrEF should be defined as a LVEF ≤ 40% [2]. All patients were covered by the affiliated health insurance of the Henry Ford Health System, the Health Alliance Plan, which allowed access to claims data. Patients were excluded if they required dialysis or supplementary oxygen. Detailed clinical information (e.g., demographics, physical examination results, past medical history, laboratory values, functional status, and medication use) and blood samples were collected at the time of enrolment. Patient deaths were identified using the Social Security Administration Death Master File, the Michigan State Division of Vital Records, and administrative data from the Henry Ford Health System through 30 April 2017. The study received approval from the Institutional Review Board of the Henry Ford Health System. All patients provided written informed consent prior to study participation.

### 2.2. Validation Dataset

Data from the “Heart Failure: A Controlled Trial Investigating Outcomes of Exercise Training” (HF-ACTION) trial [17] were used for validation. Briefly, HF-ACTION was a multicenter, randomized, controlled trial of aerobic exercise training in medically stable outpatients with HFrEF. The trial included patients with LVEF ≤ 35% and New York Heart Association (NYHA) class II to IV symptoms despite optimal medical therapy for at least 6 weeks. Patients were randomized from April 2003 to February 2007 within the United States, Canada, and France. Exclusion criteria included major comorbidities or limitations that could interfere with exercise training; recent (≤6 weeks) major cardiovascular events or planned (≤6 months) procedures; performance of regular exercise training; or use of devices that limited the ability to achieve target heart rates. The intervention consisted of 36 supervised exercise sessions followed by home-based training versus usual care alone. The primary outcome was a composite of all-cause mortality or hospitalization, and the pre-specified secondary outcomes included all-cause mortality, cardiovascular mortality or cardiovascular hospitalization, and cardiovascular mortality or heart failure hospitalization. The protocol was reviewed and approved by the appropriate institutional review board or ethics committee for each participating center and by the coordinating center’s institutional review board. The exercise intervention did not significantly affect the primary or secondary outcomes in the protocol-specified primary analysis.

### 2.3. Beta-Blocker Exposure

In the HFGR, pharmacy claims data were used to calculate beta-blocker exposure as described previously [18]. Briefly, beta-blocker doses were standardized into daily dose equivalents using doses targeted in clinical HFrEF trials or using maximum recommended daily doses if HFrEF trial data were unavailable (e.g., atenolol). The following daily doses were used for each agent: atenolol, 100 mg; bisoprolol, 10 mg; carvedilol, 50 mg; labetalol, 600 mg; and metoprolol, 200 mg (all formulations). Beta-blocker exposure was calculated over a 6-month period by multiplying the standardized daily dose equivalents by the total quantity of medication dispensed during the period, and then dividing by the total number of days during the period. By utilizing pharmacy claims data, our beta-blocker exposure values also partially account for patient adherence over an extended 6-month period. We have previously shown that this method more strongly correlates with relevant HFrEF outcomes (e.g., heart rate, hospitalization, mortality) than single time point calculations (e.g., discharge medication status) for beta-blocker exposure [19]. In HF-ACTION, beta-blocker doses were standardized into daily dose equivalents the same as in the HFGR, but only the dose at baseline (and not time-varying) was used in the analysis.

### 2.4. Genotyping

The same genotyping methods were used for both the HFGR and HF-ACTION datasets. Blood samples were processed immediately and stored at −70 °C. Genotyping was performed by the University of Michigan genotyping core lab (Ann Arbor, MI, USA) using the Axiom Biobank Genotyping Array (Affymetrix^®^; Santa Clara, CA, USA) [9]. This array includes the following ~600 K genetic variants: 1) ~300 K genome-wide variants with minor allele frequencies >1%; (2) ~250 K low frequency (<1%) coding variants from global exome sequencing projects; and 3) an additional ~50 K variants to improve African ancestry coverage (Yoruba in Ibadan, Nigeria [YRI] booster). Imputation was performed using the Michigan Imputation Server with Minimac3 [20] and the 1000 Genomes Project as the reference panel. The following filters were used for genotyping quality control: call rate < 95%, minor allele frequency (MAF) < 0.01, Hardy–Weinberg Equilibrium (HWE) *p*-value < 1.0^−8^, imputation score r^2^ > 0.8, duplicate samples identified by identity-by-state distances, and monomorphic variants.

### 2.5. Statistical Analysis

Baseline characteristics were described by mean ± standard deviation for continuous variables, and counts and percentages for categorical variables. Baseline characteristics were compared by the Student’s *t*-test, chi-square test, or Fisher’s exact test as appropriate. Proportion of YRI ancestry was estimated using ADMIXTURE software v 1.3.0. The GWAS was performed in the discovery dataset using Cox proportional hazards regression models with the primary outcome of all-cause mortality stratified by self-identified black and white race. The models consisted of each individual genetic variant, beta-blocker exposure, and the multiplicative interaction term between each genetic variant and beta-blocker exposure (genetic variant * beta-blocker exposure). The genetic variants were tested using additive genetic models. Beta-blocker exposure was included as a time-updating variable, and the models were adjusted for the following covariates: the Meta-Analysis Global Group in Chronic Heart Failure (MAGGIC) risk score (excluding beta-blocker as an input variable) [21], beta-blocker propensity score, and the first two principal components. Treatment with angiotensin inhibitors, including angiotensin-converting enzyme inhibitors and angiotensin receptor blockers, were accounted for in the MAGGIC and propensity scores. Principal components (PC) analysis was performed using EIGENSOFT [22,23]. Propensity scores for beta-blocker exposure were calculated as previously described [9,24]. Genomic control was further used to correct for test statistic inflation due to population stratification [25]. The GWAS was performed using the survival package in R, and the other statistical analyses were performed using SAS v9.4. Variants of interest were defined as having a covariate-adjusted interaction *p*-value < 5 × 10^−5^ in the HFGR. Those variants were clumped by linkage disequilibrium ([LD] r^2^ > 0.8) using PLINK and then tested for validation in similar Cox proportional hazards regression models in the HF-ACTION dataset. Variants were considered validated if the Bonferroni-corrected *p*-value was statistically significant in HF-ACTION (i.e., *p* = 0.05/# of LD-clumped variants).

## 3. Results

### 3.1. Baseline Characteristics

Baseline characteristics are displayed in Table 1. In both the discovery and validation datasets, there were fewer white than black women. In addition, white patients were older, had a higher frequency of ischemic etiology and atrial fibrillation, and had higher MAGGIC risk scores [26] compared with black patients.

In the discovery dataset only, white patients had greater LVEF, while a higher frequency of diabetes, higher systolic blood pressure and heart rate, and higher serum creatinine were observed in black patients. In the validation dataset only, white patients had a higher frequency of COPD and higher levels of NT pro-BNP, while black patients had a higher frequency of stroke, body mass index, and beta-blocker exposure. There were no significant differences in duration of follow-up and frequency of deaths between white and black patients in both datasets. The mean proportion of YRI ancestry in the self-reported black and white patients in HFGR was 83.5% and 1.0%, respectively. The mean proportion of YRI ancestry in self-reported black and white patients in HF-ACTION was 88.9% and 1.2%, respectively.

### 3.2. Discovery GWAS

The quantile–quantile (QQ) plots are displayed in Figure 1. A total of 229 variants in the black patients (Figure 1A) had interactions with beta-blocker exposure at *p* < 5 × 10^−5^ and 18 variants in the white patients (Figure 1B). The complete lists of variants with *p* < 5 × 10^−5^ and their MAF are provided in the Appendix A. Manhattan plots are presented in Figure 2 (black patients) and Figure 3 (white patients). In the black patients, one variant (rs117032090 on chromosome 7) had *p* = 1.3 × 10^−11^.

### 3.3. Validation of GWAS Results

After LD clumping, 100 of the 229 variants with *p* < 5 × 10^−5^ in the black patients and 4 of the 18 variants with *p* < 5 × 10^−5^ in the white patients remained for validation testing in HF-ACTION. None of the variants met the Bonferroni-corrected level of statistical significance in HF-ACTION, in neither the black patients (*p* < 0.0005) nor white patients (*p* < 0.0125). The interaction *p*-value for the highly significant variant rs117032090 in the HFGR was 0.9999 in HF-ACTION. One variant was suggestive for an association in the black patients: rs16844448 had an interaction *p*-value = 8.1 × 10^−6^ in HFGR and *p* = 0.005 in HF-ACTION. The intronic rs16844448 variant is in *LRP1B*, the gene encoding low-density lipoprotein (LDL) receptor-related protein 1B. Table 2 shows the results from the Cox proportional hazards regression model in black patients for rs16844448.

## 4. Discussion

To the best of our knowledge, this is the first GWAS designed to discover and validate novel genetic variants related to beta-blocker survival benefit in black and white patients with HFrEF. According to our findings, 229 and 18 variants with covariate-adjusted interaction *p*-values < 5 × 10^−5^ were identified in black and white patients, respectively. After LD clumping, one-hundred variants in black patients and four variants in white patients with *p*-values < 5 × 10^−5^ were moved forward for validation testing. None of the variants met the Bonferroni-corrected level of statistical significance in black (*p* < 0.0005) nor white patients (*p* < 0.0125) in the validation testing. However, one variant (rs16844448) was found to be suggestive of association with beta-blocker survival benefit in black patients with HFrEF.

This suggestive variant (rs16844448) is in *LRP1B*, which encodes the low-density lipoprotein (LDL) receptor-related protein-1B (LRP1B). HaploReg v4.1 shows that this intronic variant (rs16844448) alters five regulatory motifs for CEBPG, Foxa_disc5, Nkx2_11, Nkx2-4, PLZF, and RXRA_known2 [27], but its functionality has not yet otherwise been described. However, the *LRP1B* gene is located on 2q22.1, a genomic region previously implicated in premature coronary atherosclerotic disease [28,29] and heart failure [30]. Also, this suggestive variant (rs16844448) is in strong linkage disequilibrium with two other intronic variants (rs67721025 and rs70988415) in black individuals [27].

Although multiple genetic variants were identified and selected for validation, none of them were ultimately identified as a statistically significant genetic variant related to beta-blocker survival benefit in patients with HFrEF. We speculate the following reasons that could potentially have contributed to the failed validation of these genetic variants. First, the clinical characteristics in the discovery (HFGR) and validation (HF-ACTION) datasets differed. Patients enrolled in HF-ACTION had more severe HFrEF than those in HFGR, as patients in HF-ACTION had lower LVEF and higher MAGGIC risk scores. Also, exposure to beta-blockers differed between the datasets. Patients in HF-ACTION were exposed to a higher daily dose of beta-blockers than those in HFGR, not to mention that patients in the HF-ACTION clinical trial most likely had greater adherence to drug treatment than patients in the observational HFGR study. Even though we aimed to identify robust and generalizable genetic variants for different populations of patients with HFrEF, we cannot disregard those differences in disease severity and drug exposure as factors that can intrinsically modify the clinical benefit of beta-blockers in HFrEF patients.

Despite using methods such as LD clumping and *p*-value thresholding, some of the top hits identified in the discovery GWAS were considered false positive after attempted validation. These false positive variants could have somehow correlated with mortality by chance due to the high number of genetic variants tested in the GWAS. On the other hand, it is possible that there are genetic variants with smaller effect sizes that significantly interact with the survival benefit of beta-blockers in patients with HFrEF, but they were not detected in the GWAS because our study may have been underpowered. There are a few factors that could have affected the power of our study. First, the relatively small sample sizes in both the discovery and validation datasets can be explained by the challenge of enrolling patients with HFrEF, which is considered the deadliest subtype of heart failure [31], and with whole-genome data available. Second, although beta-blockers are guideline-recommended drugs with undeniable benefits for HFrEF [2], prescription filling rates for these drugs in patients with HFrEF are still considered low in the U.S. [32]. These two reasons combined contribute greatly to the fact that studies of the survival benefit of beta-blockers in HFrEF, such as ours, have small sample sizes. Moreover, stratifying the patients by black and white race further reduced the sample size, and testing for drug–gene interaction rather than the main effect further contributed to a reduction in statistical power [33].

Although this small sample size limitation exists in our study, it is noteworthy that pharmacogenomic GWASs with even smaller samples were successful in identifying and validating novel genetic variants involved in drug response. This is possible because the effect sizes of pharmacogenetic variants can be extremely large compared to the genetic variants affecting common, complex diseases. In contrast to common, complex diseases, drug responses have had less time to be affected by selective evolutionary pressures, as most drugs have only been invented in the past 100 years. As an example, the study that identified the HLA-B*5701 genotype as the determinant of flucloxacillin-induced liver injury included only 333 individuals (of which 51 were cases) and reproduced its findings in a cohort with only 23 cases [34]. The odds ratio for the HLA-B*5701 variant in this small sample was 80.6 (*p* = 9.0 × 10^−19^). Considering that the polygenic score we developed was validated in four independent datasets [35,36], this corroborates the notion that the survival benefit of beta-blockers is not only polygenic but is also formed by several small-effect genetic variants.

The interest in genomic studies that may explain the different responses to beta-blockers is not recent. In 2015, the first pharmacogenomic GWAS study failed to identify significant genetic variants that could reduce the risk of cardiovascular outcomes in patients with hypertension treated with beta-blockers [37]. However, subsequent GWASs have identified genetic variants associated with uncontrolled [38] and systolic blood pressure response [39] in patients with hypertension treated with beta-blockers. In other GWAS studies, two novel genetic variants were identified and associated with changes in heart rate in response to beta-blockers [40] as well as a genetic variant that has been associated with lower odds for new-onset diabetes in beta-blocker users [41]. To our knowledge, this is the first GWAS aimed at identifying genetic variants that predict the survival benefit from beta-blockers in HFrEF patients. Our findings could differ from these other studies mainly because the underlying mechanisms driving beta-blocker benefit in heart failure are different than in hypertension, as they do not involve lowering blood pressure [42].

Some limitations must be taken into consideration when reading this study. Low statistical power may have limited the discovery of genetic variants by the GWAS, especially for those with smaller effects. It is worth mentioning that the large clinical trials that demonstrated survival benefit of beta-blockers in heart failure date back to 20 years ago—when DNA samples from participants were not routinely collected—hindering pharmacogenomic analyses. In addition, the discovery dataset used in the GWAS was derived from a single center, and the beta-blocker exposure assessed in the validation dataset was related only to the dose at baseline—and was not time-varying. Data distinguishing potential etiologies of HFrEF other than ischemic vs. non-ischemic were not available.

## 5. Conclusions

Many variants of interest were identified in this discovery GWAS of beta-blocker survival benefit in patients with HFrEF, but none of the variants were validated in an independent dataset.

## Figures and Tables

**Figure 1 genes-14-02019-f001:**
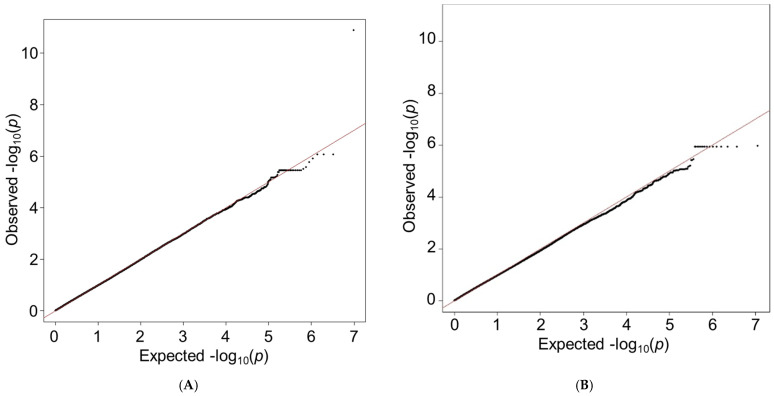
QQ plots in self-identified black patients (**A**) and white patients (**B**). The solid red lines are the *p*-values expected by chance alone. The black points are the observed *p*-values for each locus tested in the GWAS.

**Figure 2 genes-14-02019-f002:**
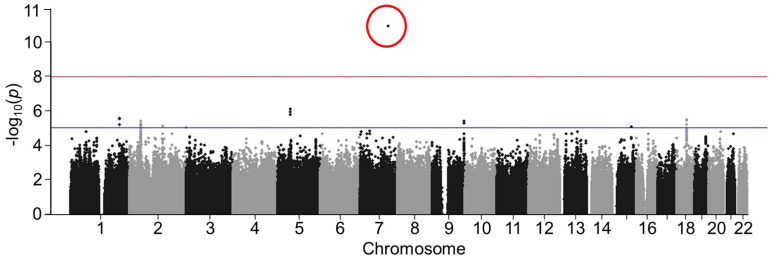
Manhattan plot in black patients. The horizontal red line indicates *p*-value = 1 × 10^−8^. The horizontal blue line indicates *p* = 5 × 10^−5^. Variants of interest were defined as *p* < 5 × 10^−5^. The red circle indicates *p* = 1.3 × 10^−11^ for rs117032090 on chromosome 7.

**Figure 3 genes-14-02019-f003:**
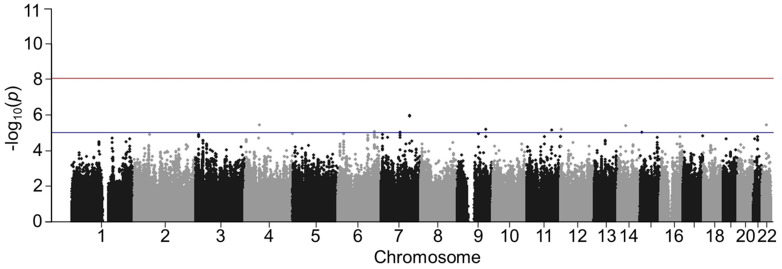
Manhattan plot in white patients. The horizontal red line indicates *p*-value = 1 × 10^−8^. The horizontal blue line indicates *p* = 5 × 10^−5^. Variants of interest were defined as *p* < 5 × 10^−5^.

**Table 1 genes-14-02019-t001:** Baseline characteristics of the discovery (HFGR) and validation (HF-ACTION) datasets stratified by self-identified race.

Characteristics	Discovery Dataset: HFGR			Validation Dataset: HF-ACTION		
	Black (*n* = 469)	White (*n* = 459)	^1^ *p*	Black (*n* = 288)	White (*n* = 579)	^1^ *p*
Female, *n* (%)	193 (41%)	141 (31%)	**0.001**	131 (45%)	137 (22%)	**<0.001**
Age, (years)	65 ± 12	71 ± 10	**<0.001**	54 ± 13	61 ± 12	**<0.001**
LVEF, (%)	33 ± 11	36 ± 10	**<0.001**	25 ± 8	25 ± 8	0.827
Ischemic etiology, *n* (%)	153 (33%)	254 (55%)	**<0.001**	89 (31%)	357 (62%)	**<0.001**
COPD, *n* (%)	105 (22%)	111 (24%)	0.569	24 (8.2%)	85 (14%)	**0.019**
Atrial fibrillation or flutter, *n* (%)	89 (19%)	169 (37%)	**<0.001**	46 (16%)	153 (25%)	**0.002**
Stroke, *n* (%)	51 (11%)	36 (7.8%)	0.141	47 (16%)	48 (7.8%)	**<0.001**
Diabetes, *n* (%)	219 (47%)	172 (37%)	**0.005**	99 (34%)	185 (30%)	0.288
Body mass index, (kg/m^2^)	31 ± 7	31 ± 7	0.140	33 ± 8	30 ± 6	**<0.001**
SBP, (mmHg)	132 ± 23	126 ± 22	**<0.001**	115 ± 18	115 ± 18	0.859
HR, (bpm)	72 ± 13	70 ± 13	**0.011**	71 ± 12	71 ± 11	0.688
NT pro-BNP, (pg/mL)	2962 ± 3260	3170 ± 3068	0.321	1266 ± 2009	1831 ± 2484	**0.002**
Serum creatinine, (mg/dL)	1.39 ± 1.08	1.18 ± 0.61	**<0.001**	1.31 ± 0.66	1.33 ± 0.76	0.675
^2^ MAGGIC risk score	17.6 ± 7.4	18.6 ± 7.0	**0.028**	18.4 ± 5.7	20.7 ± 6.6	**<0.001**
Beta-blocker exposure, (mg/day)	0.22 ± 0.32	0.22 ± 0.32	0.911	0.76 ± 0.47	0.60 ± 0.42	**<0.001**
Length of follow-up, (days)	1105 ± 695	1118 ± 707	0.780	984 ± 374	989 ± 378	0.853
Deaths, *n* (%)	112 (24%)	116 (25%)	0.677	43 (15%)	86 (15%)	>0.999

^1^ *p*-values are for the comparison between self-identified black and white patients within each dataset. Bolded *p*-values indicate *p* < 0.05. ^2^ MAGGIC risk score was calculated without beta-blockers as an input variable. COPD—chronic obstructive pulmonary disease; HF-ACTION—Heart Failure: A Controlled Trial Investigating Outcomes of Exercise Training trial [17]; HFGR—HF pharmacogenomic registry; HR—heart rate; LVEF—left ventricular ejection fraction; MAGGIC—Meta-Analysis Global Group in Chronic Heart Failure risk score [21]; NT pro-BNP—N-terminal pro b-type natriuretic peptide; SBP—systolic blood pressure.

**Table 2 genes-14-02019-t002:** Cox proportional hazards regression model results for the rs16844448 × beta-blocker exposure interaction term in black patients in HFGR and HF-ACTION.

	HFGR (*n* = 469)				HF-ACTION (*n* = 288)			
Variable	Coeff.	HR	95% CI	*p*-Value	Coeff.	HR	95% CI	*p*-Value
rs16844448 × beta-blocker exposure interaction	4.3	73.7	15.4–353.5	8.1 × 10^−6^	4.0	55.1	3.4–865.8	0.005
MAGGIC (excluding beta-blocker)	0.1	1.1	1–1.2	<0.001	0.1	1.1	1–1.2	<0.001
Beta-blocker propensity score	0.05	1.0	0.9–1.2	0.600	0.04	1.0	0.8–1.4	0.800

CI—confidence interval; Coeff.—coefficient; HF-ACTION—Heart Failure: A Controlled Trial Investigating Outcomes of Exercise Training [17]; HFGR—Heart failure pharmacogenomic registry [9]; HR—hazard ratio; MAGGIC—Meta-Analysis Global Group in Chronic Heart Failure risk score [21].

## Data Availability

The data from HFGR are publicly available via the database of Genotypes and Phenotypes (https://www-ncbi-nlm-nih-gov.proxy.lib.umich.edu/gap (accessed on 23 October 2023). The remaining data that support the findings of this study may be made available upon reasonable request from qualified researchers trained in human subject confidentiality protocols by contacting D.E.L. (dlanfea1@hfhs.org) at Henry Ford Hospital.

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
