# Peer review of "Genome-Wide Association Study of Beta-Blocker Survival Benefit in Black and White Patients with Heart Failure with Reduced Ejection Fraction"

_genes, 2023, doi:10.3390/genes14112019_

Round 1
Reviewer 1 Report
Comments and Suggestions for Authors
Luzum, et al. performed a GWAS to identify genetic variants associated with beta-blocker survival benefit in HFrEF patients. Overall, the study addresses an important research question, and the manuscript is well written. I have a few comments:
· - The authors have previously developed and validated a PRS for beta-blocker benefit in White patients, suggesting that survival benefit of beta-blockers is polygenic. What was the rationale for performing a GWAS and not attempting to develop a PRS for Black patients in the current study?
· - Did the authors adjust the analyses for any concomitant HF therapies that might also impact survival, such as isosorbide dinitrate/hydralazine?
· - The sentence “The protocol was reviewed and approved….” is repeated on lines 100-103 and 107-109.
Reviewer 2 Report
Comments and Suggestions for Authors
The manuscript describes a GWAS aimed at identifying genetic markers associated with influencing beta-blocker survival benefit in patients with reduced ejection fraction heart failure (HFrEF). Interactions between gene variants and beta-blocker exposure from baseline were investigated with Cox proportional hazards models adjusted for clinical risk and propensity scores.
A discovery a self-identified race-stratified cohort of 469 black and 459 white patients with HFrEF was used for discovery and yielded a total of 229 and 18 variants interactions with beta-blocker exposure at p<5 x 10-5 in black and white HFrEF patients, respectively. After LD-pruning, 100 and 4 variants in the black and white patients, respectively, were tested for validation in a multicenter HFrEF clinical trial (HF-ACTION, n=288 black and 579 white patients). However none of the variants validated in the independent validation dataset at a Bonferroni-corrected level of statistical significance, in either the black patients (p < 0.0005) or white patients (p < 0.0125). One variant was suggestive for an association in the black patients, rs16844448, with an interaction p-value = 8.1 x 10-6 in the discovery HFGR and p = 0.005 in the validation HF-ACTION study. rs16844448 is an intronic variant in LRP1B, the gene encoding low density lipoprotein (LDL) receptor related protein 1B.
General comments
The manuscript is very well written and the authors have a strong track record in this field of research. The design of the study is appropriate. As the authors note a smaller than ideal sample size limitation exists in their study, but it is not impossible to identify pharmacogenetic variants in studies of this type and size.
Some expansion of some details of the manuscript would improve it, as outlined below.
Specific Comments
Table 1 – inclusion of other aetiologies besides ischaemic would be informative.
The fonts of the axis labels for all figures should be enlarged.
The relevance of the horizontal lines at -log10(P) 5 and 8 in the Manhattan plots should be stated in the figure legends.
Figure 2 – the y-axis scale maximum should be increased to encompass the maximum data point of p = 1.3 x 10-11 for rs117032090 on chromosome 7.
Also circling this datapoint in another colour would make it stand out.
How was LD pruning performed?
A table with a Cox proportional hazards model of the best performing variant in the validation study, rs16844448, and its interaction with beta-blocker exposure along with the study’s stated covariates: the Meta-Analysis Global Group in Chronic Heart Failure risk score (excluding beta-blocker), beta-blocker propensity score, and the first two principal components, would be an informative addition to the Results section.
A comment on some estimate of the degree of genetic admixture of the stratified race groups in the HFGR and HF-ACTION study cohorts in the Discussion section would be enlightening.
